# Effect of Neuro-Adaptive Electrostimulation Therapy versus Sham for Refractory Urge Urinary Incontinence Due to Overactive Bladder: A Randomized Single-Blinded Trial

**DOI:** 10.3390/jcm12030759

**Published:** 2023-01-18

**Authors:** Álvaro Zapico, Julia Ercilla, Javier C. Angulo, Vicente Pérez, Juan Nicolás Cuenca, Diana Barreira-Hernández, Carlos Udina-Cortés

**Affiliations:** 1Department of Obstetrics and Gynecology, Hospital Universitario Príncipe de Asturias, 28802 Alcalá de Henares, Spain; 2Department of Surgery, Universidad de Alcalá, 28802 Alcalá de Henares, Spain; 3Department of Urology, Hospital Universitario de Getafe, 28905 Getafe, Spain; 4Clinical Department, Facultad de Ciencias Biomédicas, Universidad Europea, 28675 Villaviciosa de Odón, Spain; 5Nursing and Physiotherapy Department, Universidad de Alcalá, 28802 Alcalá de Henares, Spain; 6Department of Biomedical Sciences, Universidad de Alcalá, 28802 Alcalá de Henares, Spain; 7Institute Neurolife, 28034 Madrid, Spain

**Keywords:** overactive bladder, urinary incontinence, neuromodulation, neurostimulation, neuro-adaptive therapy, SCENAR

## Abstract

This randomized clinical trial evaluates the success rate of neuro-adaptive therapy (NAT), applied with a specific neuro-adaptive regulator device, the Self-Controlled Electro Neuro-Adaptive Regulation (SCENAR), versus a sham for urge incontinence due to an overactive bladder (OAB). From February 2019 to May 2021, 66 patients were recruited. All subjects were randomized 1:1 at the first intervention visit to the NAT or sham procedure. Inclusion criteria were females between 18 and 80 years old with leakages due to an overactive bladder with unresponsiveness to medical therapy. Subjects were scheduled to receive up to eight weekly 20 min intervention sessions to obtain a complete (CR) or partial response (PR). Patients with no response after three sessions were considered as a failure. The primary end point of this trial was to assess the efficacy of NAT compared to an inactive sham intervention, evaluated 1 month after the last session. Analysis showed 23 (70%) patients responded (20 complete and 3 partial response) in the NAT group compared to 16 (48%) patients (all complete response) in the placebo arm (*p* = 0.014). Significant differences were maintained after the intervention, with persistent response at 3 months in 19 (58%) patients after active treatment and 14 (42%) after the placebo (*p* < 0.001), and at 6 months in 18 (55%) vs. 11 (33%) (*p* = 0.022), respectively. The number of sessions to achieve CR was similar in both arms, with 4.3 ± 1.9 in NAT and 3.9 ± 1.8 in the sham group (NS). Significant differences were observed between both groups for patients’ satisfaction (*p* = 0.01). The binary model selected age as a predictor of response at the last follow-up. The odds ratio indicates that each year of increase in age, the probability of a positive response to treatment at 6 months decreases 0.95 (95% CI 0.9–0.99) times (*p* = 0.03). In conclusion, this pilot randomized trial gives evidence that neuro-adaptive electrostimulation is effective to treat refractory urge urinary incontinence due to OAB. The security and long-term efficacy of this treatment merits further evaluation. Moreover, its favorable profile and the economic advantages of the device make the evaluation of this promising technique mandatory in a primary therapeutic scenario.

## 1. Introduction

Overactive bladder (OAB) is characterized by the storage symptoms of urgency with or without incontinence, usually with frequency and nocturia [1]. The etiology of OAB is probably multifactorial, including changes in anatomy and body composition, lifestyle factors, and comorbidities. Overactive bladder syndrome impacts on quality of life that may lead to social isolation [2]. Initial management consists of behavioral therapy. Following behavioral and pelvic floor therapies, drug therapies are the mainstay of treatment [3]. Selective beta-3 agonist mirabegron tends to be used as a first-line medical therapy for a better efficacy/tolerability balance [4,5]. However, limited efficacy, side effects, and costs result in a short adherence to this therapy. Antimuscarinic drugs are also widely used, though they are frequently discontinued due to low efficacy and bothersome side effects [6]. The prevalence of OAB in Spain is approximately 20% in patients ≥40 years of age and could be much more frequent if actively screened [7,8]. An increasing OAB incidence is expected according to the aging baby boomer population in western countries [9].

Non-drug active therapies have been developed as serious alternatives for non-neurogenic OAB syndrome [10]. Worldwide, neuromodulation is considered as a third-line procedure when appropriate lifestyle advice, physical therapies, and medication fail [11]. At present, the decision on which third-line therapy to perform is based on the clinicians and patient preferences and there is no evidence-based hierarchy available for guidance. The US Food and Drug Administration (FDA) approved sacral neuromodulation (SNM) in 1997 for OAB syndrome [12]. SNM requires a permanent surgical implant placed at the sacral foramen. The therapeutic improvement or cure rate ranges from 61 to 90% [13,14]. A complication rate of up to 40% over 5 years (range 15–42%) has been reported [12]. Most of these complications have been adverse events related to pain at the implant site and undesirable changes in stimulation [13,14,15,16].

Percutaneous tibial nerve stimulation (PTNS) was FDA approved in 2005 for non-neurogenic lower urinary tract dysfunction [17]. Therapeutic improvement or cure range from 54 to 79% [18,19,20]. No serious device-related adverse events or malfunctions have been reported. For PTNS, a needle electrode is inserted 5 cm cephalad to the medial malleolus and a surface electrode is placed on the ipsilateral calcaneus. The standard approach requires 12 weekly 30 min intervention sessions. Several studies give evidence that electrical stimulation may be better than anticholinergics [10], suggesting that non-implantable electrical stimulation might be an option in patients’ refractory OAB that merits further investigation.

Parasympathetic nervous signal damping using the adaptive neuro-fuzzy inference system method to control an overactive bladder is under investigation [21]. We report the outcomes of a single center clinical trial to assess the efficacy of a new neuromodulation technique, neuro-adaptive therapy (NAT), compared to a validated sham intervention in females with OAB and urge urinary incontinence refractory to medical therapy. This randomized trial is the first one to be reported about the efficacy of NAT on OAB patients.

## 2. Materials and Methods

### 2.1. Study Design

NAT is applied using a specific neuro-adaptive regulation device named SCENAR (Urogyne version). This trial, registered as SCENAR-EC trial (NCT04164589), is an institutional review board approved, randomized controlled single-blinded trial conducted with consecutive patients treated in an academic institution.

From February 2019 to May 2021, 66 patients were recruited. All subjects were randomized 1:1 at the first intervention visit to NAT or sham using a random block design stratified by investigational site.

Patients were screened for the trial in a urogynecology office of a tertiary care university hospital in which clinical history, physical examination, bladder diary, and self-assessed questionnaires ICIQ-SF and Sandvik were collected. Patient selection and enrollment, and the allocation of treatment was performed by two experienced gynecologists (A.Z. and J.E.).

The inclusion criteria were females between 18 and 80 years old with urge urinary incontinence due to OAB, with failed conservative care, unresponsive to drug therapy, or in whom side effects forced them to stop the medication. Self-reported duration of symptoms was always more than 3 months. Previous medications (antimuscarinics or beta-3 agonist) were discontinued more than 4 weeks before inclusion in all cases. Mixed incontinence was not an exclusion criterion, provided that the most bothersome symptom was urge incontinence. All patients gave informed consent and were capable of ambulatory care and able to use the toilet independently. The exclusion criteria were pregnancy, use of pacemaker, neurogenic bladder dysfunction, current urinary tract or vaginal infections, onabotulinumtoxinA use within the last year and previous PTNS or SNM treatment.

To fulfill all the aforementioned criteria, patients were clinically evaluated prior to trial inclusion. Clinical evaluation included pelvic examination to exclude gynecologic factors such as significant pelvic organ prolapse (III–IV/IV). The Oxford test and stress test (cough and strain) were carried out to detect leakage likely due to sphincter incompetence or urethral hypermobility. Urine culture and ultrasound were used to exclude infection or obstructive origin. Urodynamic testing was only conducted in those cases of mixed incontinence whereas a reasonable doubt of predominant component existed, to confirm detrusor overactivity.

Patients were informed to register, in a specific designed form, voiding diary parameters including frequency, night-time voids, urgency for each voiding, and urge urinary incontinence episodes. This self-registered process started at least 1 week before the first intervention session and continued until finishing all the treatment sessions. At first intervention day, the self-administered International Consultation on Incontinence Questionnaire-Short Form (ICIQ-SF) quality scores [22,23] and severity score (Sandvik) [24] were recorded in a designed database. Both instruments are validated for the Spanish population. ICIQ-SF score range from 0 to 21, with 11 points that score frequency and severity of incontinence, and 10 points that evaluate the impact on quality of life (QoL). The Sandvik test range from 0 to 12 and measured urine leakage is based upon self-registered voiding diary parameters, but does not consider QoL evaluation.

Subjects were scheduled to receive up to 8 weekly 20 min intervention sessions to obtain complete or partial response. Patients with no response after 3 sessions were considered as a failure. A complete response was considered when Sandvik and ICIQ-SF scores reached 0. A partial response was considered when achieving 50% or more reduction of Sandvik and ICIQ-SF scores compared to baseline. Intervention sessions were stopped when there was a failure or complete response. Patients with a partial response continued treatment sessions until reaching a complete response or up to a maximum of 8 sessions. Follow-up was scheduled at 1, 3, and 6 months after the last intervention session, to evaluate the response according to the same data recorded on the treatment sessions. Therefore, patients were informed to register voiding diary parameters at least one week before the follow-up scheduled date to be able to self-register the questionnaires. Patient satisfaction with the procedure was evaluated using a self-assessed Likert scale (range from 0 to 10).

### 2.2. Outcomes Evaluated

The primary endpoint of this trial was to assess the efficacy (response rate) of NAT compared to an inactive sham intervention in women with OAB symptoms in an intent to treat analyses. Primary evaluation was performed 1 month after the last intervention session. Secondary endpoints included the evaluation of the response rate at 3 and 6 months of follow-up after the last intervention session. The comparison of the number of sessions needed to achieve a complete response was also evaluated. Self-reported patient satisfaction with the procedure was also evaluated at 1, 3, and 6 months of follow-up. Outcomes were evaluated by the two experienced urogynecologists responsible (A.Z. and J.E.).

### 2.3. Procedures and Intervention

A device model SCENAR^®^ 1NT-01.C version Urogyne (Ritm OKB ZAO, Taganrog, Russia) with non-invasive external reusable electrodes was used for vulvoperineal and sacral treatment. It is a transdermal neurostimulator for the non-invasive treatment of pain and physiological systems disturbance.

NAT was applied through non-invasive reusable skin surface electrodes. During the intervention sessions, subjects were first in the lithotomy position for a vulvoperineal approach and later in a standing up position for sacral neuromodulation. The procedure started testing the energy setting on the medial aspect of the thigh. For patients randomized to the NAT group, energy was set at a level that could be felt comfortably by the subject, then this was reduced until the maximum level where it could not be felt. In the sham group, energy was set at a level that could be felt by the subjects and then it was reduced to 0 and the electrode was disconnected. The audible sound produced by the SCENAR device was canceled in both groups so that subject could not receive any auditory variation.

Treatment was started by applying NAT at the peripheral level on the vulvoperineal area at S2–S4 dermatome skin projections and completed by direct modulation on S2–S4 sacral foramens through a posterior approach. All treatments were administered by an experienced nurse (V.P.) with the same equipment.

### 2.4. Statistical Analysis

Statistical analysis was performed with the R Ver.3.5.1 program (R Foundation for Statistical Computing, Institute for Statistics and Mathematics, Welthandelsplatz 1, 1020 Vienna, Austria). Missing data on outcome variables were analyzed by intention to treat. The level of significance was established at *p* < 0.05. The distribution of the quantitative variables was tested with the Shapiro–Wilk test, which showed the absence of normality. Quantitative variables are shown with mean ± SD and qualitative variables with absolute and relative values (%). The presence of significant baseline differences between both groups was tested using Fisher’s exact test for qualitative variables and the Mann–Whitney U test for quantitative variables.

Quantitative outcome variables were analyzed using a robust repeated measures model with two factors, between (group) and within (measurements) on the 20% trimmed means, due to their non-normal distribution; the omnibus test reports a robust ANOVA at its level of significance. For the post hoc tests, the intra-group Wilcoxon Signed Rank test and the Mann–Whitney U test between groups were applied, applying a Bonferroni correction in both cases. Qualitative variables were analyzed with the Cochran–Mantel–Haenszel test after fulfilling the assumption of homogeneity of the odds ratio using the Wolf test, while for the post hoc Fisher’s exact test was applied with Bonferroni correction.

A logistic regression model was applied between the dependent variable response at 6 months with a binomial model and the explanatory variables age, body mass index, previous pelvic surgery or disorder, and baseline Sandvik.

## 3. Results

### 3.1. Clinical Data

Sixty-six consecutive patients were recruited in the trial. All suffered refractory urge incontinence due to OAB bladder. Fifty-four (81.8%) were unresponsive to medical treatment and 12 (18.2%) non-tolerant due to side effects. They were randomized to receive either active or sham neuro-adaptive electrostimulation.

Figure 1 shows the study flow chart. Overall, 102 patients were enrolled and after the exclusion of 36 patients, a total of 66 patients were randomized, and 33 were allocated to each treatment arm. Some patients meeting the inclusion criteria did not consent to the restrictions caused by the COVID-19 pandemic lockdown at the start of the trial. Table 1 presents the clinical characteristics of the patients in each group. There was no statistically significant difference between treatment groups regarding age, anthropomorphic parameters, previous pelvic surgery, and accompanying stress urinary incontinence. In addition, the number of urgency and urge incontinence episodes, micturition during night-time, and both ICIQ-SF and Sandvik scores were equivalent between treatment arms.

Three patients in the NAT group were lost during the treatment sessions as they abandoned treatment due to fear of COVID-19 contact during the pandemic restrictions. One was an 80-year-old patient who quit after the fifth session with a Sandvik test change from 8 at baseline to 3 at the fifth session. The second patient who abandoned the study was a 76-year-old female who stopped treatment after the third sessions with a Sandvik test change from 6 to 2. The last drop-out was an 80-year-old patient who also withdraw consent for fear of COVID-19 limited access to the hospital. There was no abandonment in the placebo group. None of the patients receiving the active treatment or placebo complained of any treatment-related complications.

### 3.2. Response to Treatment

Primary endpoint analysis showed 23 (70%) patients in the ITT population of the active treatment group responded to neuro-adaptive electrostimulation as scheduled, compared to 16 (48%) patients in the sham group (*p* = 0.014). In the active treatment arm, a partial response was observed in 3 cases and complete response in the remaining 20 patients while all patients in the control group showed a complete response. The total number of treatment sessions received was 6 ± 1.50 in the NAT group and 5.4 ± 1.4 in the sham group (NS). A partial response was achieved after 2.4 ± 1.3 sessions in the NAT group vs. 1.9 ± 0.8 in the sham group (NS). The number of sessions for a complete response was also similar in both groups, 4.3 ± 1.9 vs. 3.93 ± 1.8 (NS), respectively. The therapeutic procedure was well tolerated and no treatment-related complications were seen either during treatment or during the follow-up.

Table 2 shows the study outcomes at each follow-up visit, thus including the evaluation of treatment response 1 month after intervention already described (primary endpoint) and also within follow-up (secondary endpoints). The number of patients actively treated in the NAT group that showed response was 23 (70%), 19 (58%) and 18 (55%) vs. 15 (45%), 14 (42%) and 11 (33%) in sham group at 1, 3 and 6 months, respectively. The difference was statistically significant at all time values (*p* = 0.014 at month 1; *p* < 0.001 at month 3; *p* = 0.022 at month 6). Failure rate increased with follow-up in both treatment groups. Differences of statistical significance in the occurrence of recurrences between the two treatment groups were not evidenced with log-rank (*p* = 0.683) or Wilcoxon–Peto (*p* = 0.236) models.

### 3.3. Self-Administered Questionnaires

Figure 2 represents the evolution of the ICIQ-SF and Sandvik score baseline and after treatment at 1, 3, and 6 months of follow-up, with a significant group effect detected for ICIQ-SF (*p* = 0.002) but not for Sandvik score (*p* = 0.07).

Table 3 shows the evolution of the scores evaluated in the questionaries during the study. Table 4 specifies the group effect, the time effect, and the group–time interactions. For ICIQ-SF, the ANOVA revealed a highly significant effect of time (*p* < 0.001), and for the group-by-time interaction approached statistical significance (*p* = 0.06). For severity Sandvik score the ANOVA revealed a significant effect for both time (*p* < 0.001) and for the group-by-time interaction (*p* = 0.049).

### 3.4. Self-Assessed Satisfaction with the Procedure

Table 3 and Table 4 confirm differences in self-reported patient satisfaction with the procedure at all times evaluated, another secondary endpoint of the trial. The group effect was highly significant (*p* < 0.001) and significant differences were found between groups over time (*p* = 0.016). A group-by-time interaction effect was also confirmed (*p* = 0.01), thus confirming the difference between groups is different at different times.

### 3.5. Prediction of 6-Months Response

A logistic regression binomial model was applied between the dependent variable response at 6 months and explanatory variables including age, body mass index, previous pelvic surgery, and baseline Sandvik. The binary model selected the variable age as a predictor. The odds ratio indicates that with each year increment in patient age, the probability of a positive response to treatment at 6 months decreases 0.952 (95% CI 0.9–0.994) times (*p* = 0.03). Sensitivity and specificity are around 69.2% with a cut-off point of 45.6%. The ROC curve shows a significant AUC of 67% (95% CI 53.4–67).

## 4. Discussion

This study showed evidence of the therapeutic effect of NAT applied with the specific neuro-adaptive regulator device SCENAR on overactive bladder syndrome. In spite of the high response to the placebo observed in this trial, primary end point evaluation showed that neuro-adaptive therapy was significantly better (70 vs. 48%). Secondary endpoints, such as the evaluation of response rate at 3 and 6 months of the completion of the procedure and self-reported patient satisfaction with the procedure at 1, 3, and 6 months of follow-up have been fulfilled as well. That means the clinical improvement observed in active treatment is maintained after 6 months with a difference between group scoring of 2.2 (0.6–5) on the ICIQ-SF questionnaire, despite the response in the sham group being higher than expected and higher than reported in other neuromodulation trials [12].

Both SNM and PTNS have been confirmed to be safe and effective treatments for OAB symptoms, with an overall success or improvement rate ranging from 61 to 90% for SNM [13,14,15,16] and from 54 to 79% for PTNS [18,19,20,25]. Most of these studies use medical therapy as the control group, and few of them are based on sham stimulation as the control [19,20], a better method to properly assess the effectiveness of the intervention. Conversely, in uncontrolled studies, the placebo effect cannot be evaluated out of the therapeutic effect. Our study used a sham approach for the control group. Another interesting peculiarity of the population to which our treatment was offered is that they were patients refractory to OAB oral-drug medical therapy with, at least theoretically, more limited expected response. Despite that, the objectives raised were satisfactorily fulfilled. The simplicity of the treatment used, the short learning curve, and the absence of side effects may favor dissemination of the procedure at different health assistance levels.

The handheld electro-therapeutic device we used has an integrated electrode that delivers computed-modulated electrostimulation via the patients’ skin and monitors the skin impedance constantly, thus providing a biofeedback mechanism. However, what is more attractive in the procedure we used is that it is technically very simple when compared to neuromodulation and/or tibial nerve stimulation. The permanent implant of SNM is not needed and a lower number of therapeutic sessions are needed to achieve response than those reported in PTNS protocols.

Neuro-adaptive stimulation therapy serves a double purpose by combining the effects of SNM and PTNS. Peripheric neuromodulation is applied on vulvar dermatome S2–S4 cutaneous projections and a central one is applied directly on metameric S2–S4 sacral roots. Another valuable advantage of the procedure is that it uses non-invasive surface reusable electrodes that imply a very low health cost.

The severity Sandvik score test showed significant differences from baseline in the neuro-adaptive group but not in the control group, but the group-by-time interaction effect using this score confirmed the difference between groups is different at different times. Conversely, statistical differences between both groups were overtly confirmed by the ICIQ-SF questionnaire. This score test ranges from 0 to 21, with 10 points depending on the self-scoring of quality of life disturbance. On the other side, the Sandvik test is based on recording the measured urine leakage with no scoring on QoL. This difference could stand in a slightly different performance of both tests, possibly based on the patient’s fear to a relapse.

OAB is a complex condition that compromises QoL but does not affect survival. So, the potential benefit of every treatment should be carefully weighed against adverse events. Based on that, treatment alternatives are classified as first-, second-, or third-line according to an expected benefit/risk evaluation [3]. Studies of refractory urge urinary incontinence have demonstrated efficacy for both onabotulinumtoxinA and SNM as third-line procedures, but treatment-related adverse effects include urinary infection, the need for clean intermittent self-catheterization and revision or removal of SNM devices [14,26]. Based on our experience neither complications nor side effects were seen with neuro-adaptative stimulation and according to the technical profile of the procedure, they are not expected.

The median time to discontinuation of medical treatment is quite short in the Spanish population, from 56 to 90 days when antimuscarinic or mirabegron are used. At 12 months the adherence of oral medical treatments is only 10.2–20.2% [9]. Considering this scenario, together with the simplicity of the neuro-adaptive therapeutic procedure evaluated and the results presented here on OAB refractory to medical treatment, we consider that NAT should also be investigated in the primary first-line setting, especially for patients unwilling to undergo medical treatment or with a low-adherence to drugs. Alternatively, another interesting source of evidence will be the comparison of neuro-adaptive stimulation to third-line treatments including onabotulinumtoxinA or other modalities of invasive neuromodulation.

Neuro-adaptive electrostimulation therapy to treat pain and related entities has been extensively studied and used in Russia for several decades, but lacks the standards required for international scientific reporting. Moreover, there is no previous experience with this technique regarding overactive bladder treatment and urge incontinence. For these reasons, we decided to register and perform this clinical trial. Our pilot experience presents promising results, but we must acknowledge important limitations to our study. Low-frequency modulated electric current therapy has been recently evaluated for pain relief secondary to osteoarthritis and fibromyalgia [27,28,29] and, despite there being growing evidence of the therapeutic basis for NAT through parasympathetic nervous signal damping, the clinical experience with this form of therapy is still very limited. As far as we know, this is the first clinical trial conducted with the indication of urge incontinence due to OAB. We were unable to identify markers of response other than age, otherwise a well-known prognostic factor for OAB using different treatment modalities [26]. Additional studies are needed to confirm our promising results and to evaluate the long-term efficacy and safety of NAT in this therapeutic use. In addition, further investigation should be performed to investigate neuro-adaptive stimulation in different subgroups of patients with OAB, such as those with coincident stress urinary incontinence, specific age groups, and also the male population. Finally, despite the procedure being technically very simple, its standardization and reproducibility merit further investigation.

## 5. Conclusions

Neuro-adaptive stimulation therapy is a promising approach to treat urge incontinence in patients with OAB. It achieves better response rate than placebo in cases refractory to oral medication. Validated questionnaire scores for ICIQ-SF and Sandvik tests consistently improve and continued efficacy is confirmed up to 6 months after treatment. Moreover, patient reported outcomes based on self-assessed satisfaction also confirm the efficacy of this approach compared to a placebo.

## Figures and Tables

**Figure 1 jcm-12-00759-f001:**
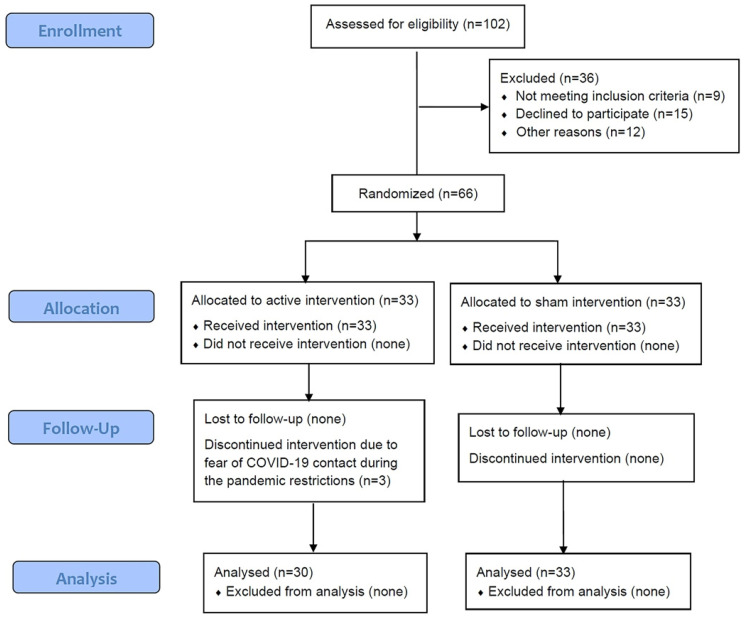
Study flow diagram, according to CONSORT 2010 for transparent reporting of trials.

**Figure 2 jcm-12-00759-f002:**
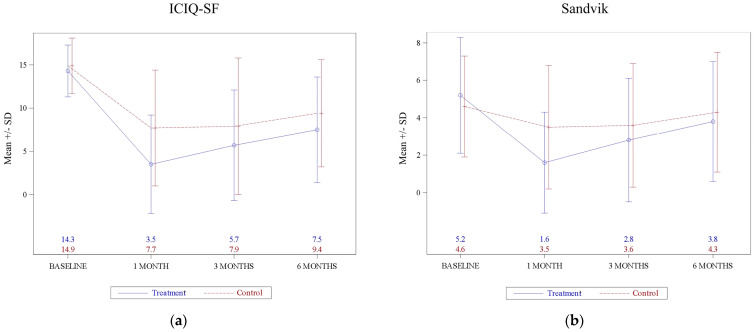
ICIQ-SF and Sandvik score evolution for active treatment and placebo during treatment and follow-up: (**a**) ICIQ-SF; (**b**) Sandvik.

**Table 1 jcm-12-00759-t001:** Demographic and clinical characteristics among study participants at baseline.

Variables	Active (n = 33)	Sham (n = 33)	*p* Value
Age (year)	61.4 ± 11.4	60.9 ± 11.9	0.89
Weight (kg)	73.3 ± 14.5	69.7 ± 11.9	0.41
Height (cm)	160.1 ± 6.1	161.7 ± 4.9	0.21
Body mass index	28.5 ± 5.3	26.4 ± 3.8	0.08
Previous pelvic surgery, n (%)	8 (24)	8 (24)	0.99
Stress incontinence ^1^, n (%)	4 (12)	7 (21)	0.5
Urge incontinence episodes ^2^, n	1.7 ± 1.7	1.6 ± 1.9	0.45
Urgency episodes ^2^, n	2.1 ± 1.9	2.3 ± 1.9	0.56
Nocturia ^3^, n	1.8 ± 1.5	2 ± 1.2	0.87
ICIQ-SF score	14.4 ± 3	14.9 ± 3.2	0.68
Sandvik score	5.2 ± 3.1	4.6 ± 2.7	0.45

^1^ Urge incontinence predominant; ^2^ Per day; ^3^ During night-sleep. Data expressed with mean ± SD or with absolute and relative values.

**Table 2 jcm-12-00759-t002:** Outcomes for each treatment group at all study follow-up visits, ITT population. Patients responding, treatment failures, and patients who abandoned the study are specified.

Outcome	Active NAT Group, n (%)			Sham Group, n (%)		
	Month 1	Month 3	Month 6	Month 1	Month 3	Month 6
Response #	23 (70) *	19 (58) **	18 (55) ***	16 (48) *	14 (42) **	11 (33) ***
Complete Partial	20	15	9	16	11	8
	3	4	9	-	3	3
Failure	7 (21)	11 (33)	12 (36)	17 (52)	19 (58)	22 (67)
Abandoned	3 (9)	3 (9)	3 (9)	-	-	-

# Comparison of response rate (ITT) during follow-up: *p* value = 0.014 *; *p* value < 0.001 **; *p* value = 0.022 ***.

**Table 3 jcm-12-00759-t003:** Mean ± SD values obtained by self-assessed questionnaires performed during the study.

Questionnaire	Treatment Group	Baseline	Month 1	Month 3	Month 6	Dif. * (95% CI)
ICIQ-SF (0–21)	Active	14.4 ± 3	3.5 ± 5.7	5.7 ± 6.4	7.5 ± 6.1	2.2 (0.6, 5.0)
	Sham	14.9 ± 3.2	7.7 ± 6.7	7.9 ± 6.7	9.5 ± 6.2	
Sandvik (0–12)	Active	5.2 ± 3.1	1.6 ± 2.7	2.8 ± 3.3	3.8 ± 3.2	0.6 (−0.9, 2.1)
	Sham	4.6 ± 2.7	3.5 ± 3.3	3.6 ± 3.3	4.3 ± 3.2	
Satisfaction (0–10)	Active	-	8.4 ± 2.5	7.2 ± 3.1	6.5 ± 2.8	−1 (−2.2, 0.2)
	Sham	-	5.8 ± 3.1	6.1 ± 3.2	5.5 ± 2.8	

* 6 month difference between groups (95% CI).

**Table 4 jcm-12-00759-t004:** Effects of treatment received and their *p* values for self-assessed questionnaires.

Questionnaire	Group Effect	Time Effect	Group–Time Effect
ICIQ-SF (0–21)	0.002	<0.001	0.062
Sandvik (0–12)	0.072	<0.001	0.049
Satisfaction (0–10)	<0.001	0.016	0.01

## Data Availability

Full data will be provided by the main investigator of the trial upon reasonable request.

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
