# Peer review of "Effect of Neuro-Adaptive Electrostimulation Therapy versus Sham for Refractory Urge Urinary Incontinence Due to Overactive Bladder: A Randomized Single-Blinded Trial"

_jcm, 2023, doi:10.3390/jcm12030759_

Round 1
Reviewer 1 Report
1. The patient selection and inclusion criteria must be given more detailed. Did they use a valid questionnaire to diagnose urge incontinence?
2. Who performed the initial examination and pelvic examination? Were they the same clinicians?
3. Including patients who also have stress-incontinence might affect results. I suggest to exclude those patients and make the statistics again.
Reviewer 2 Report
In general, the publication is good, and the method deserves to be studied and presented to a Western audience. However, it seems strange that more than 30 years of successful clinical experience in Russia is ignored. This information is available from the manufacturers of the device.
Round 2
Reviewer 1 Report
Having similar heterogenous two groups does not constitute a homogenous group.